# Image-computable Bayesian model for 3D motion estimation with natural stimuli explains human biases

**Daniel Herrera-Esposito**
Department of Psychology
University of Pennsylvania
Philadelphia, Pennsylvania 19104
`dherresp@sas.upenn.edu`

**Johannes Burge**
Department of Psychology
University of Pennsylvania
Philadelphia, Pennsylvania 19104
`jburge@psych.upenn.edu`

## Abstract

Estimating the motion of objects in depth is important for behavior, and is strongly supported by binocular visual cues. To understand how the brain should estimate motion in depth, we develop image-computable ideal observer models from naturalistic binocular video clips of two 3D motion tasks. The observers spatio-temporally filter the videos, and non-linearly decode 3D motion from the filter responses. The optimal filters and decoder are dictated by the task-relevant statistics, and are specific to each task. Multiple findings emerge. First, two distinct filter types are spontaneously learned for each task. For 3D speed estimation, filters emerge for processing either changing disparities over time (CDOT) or interocular velocity differences (IOVD), cues used by humans. For 3D direction estimation, filters emerge for discriminating either left-right or towards-away motion. Second, the covariance of the filter responses carries the information about the task-relevant latent variable and the filter responses, conditioned on the latent variable, are well-described as jointly Gaussian. Quadratic combination is thus necessary for optimal decoding. Finally, the ideal observer yields non-obvious, counter-intuitive patterns of performance like those exhibited by humans. Important characteristics of human 3D motion processing and estimation may therefore result from optimal information processing in the early visual system.

## 1 Introduction

Estimation of motion in depth, generated by self-motion and by the motion of objects in the world, is important for successful interaction with the environment. Animals with binocular vision combine 2D images formed by each eye to estimate 3D motion. However, it is not well understood how the brain does, and should, estimate 3D motion from binocular images [12]. Ideal observer analysis is a useful tool for addressing such questions [6, 15].

To better understand the computations that optimize local 3D motion estimation, we developed two image-computable Bayesian ideal observer models that were constrained by the front-end of the human visual system and matched to the statistics of naturalistic binocular video clips. Each ideal observer was designed to solve one of two specific tasks: 3D speed estimation or 3D direction estimation. For each task, the observer model first applies a small set of spatio-temporal filters to the binocular videos. Then, using the statistics of the filter responses and the tools of probabilistic inference, it performs non-linear decoding to yield optimal estimates of the task-relevant latent variable (i.e. 3D speed or 3D direction). The filters are learned via Accuracy Maximization Analysis (AMA), a filter-learning approach designed to find the stimulus features that provide the most useful information for a given task [9, 16]. Here, we use AMA-Gauss [20], a computationally efficient

4th Workshop on Shared Visual Representations in Human and Machine Visual Intelligence (SVRHM) at the Neural Information Processing Systems (NeurIPS) conference 2022. New Orleans.

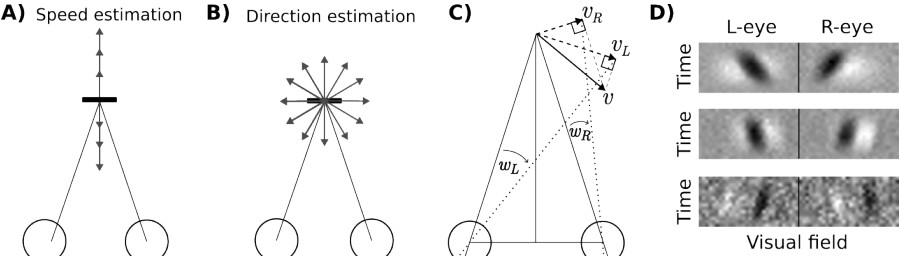

Figure 1: **Task design and stimuli. A)** Top-down view of the observer's eyes (circles) and the target surface (horizontal black bar) for the speed estimation task. Gray arrows show the types of 3D motions used at this task. **B)** Same as **A)**, but for the direction estimation task. **C)** Projective geometry linking 3D motion ($v$) to retinal speeds ($w_L$, $w_R$) in degrees of visual angle per second. **D)** Three example binocular videos from the speed estimation dataset, with different underlying speeds.

form of AMA that is empirically justified (see below) for the 3D motion tasks being investigated here. For each task, two filter types with distinct functional specializations are learned, reflecting a spontaneously developed division of labor.

The ideal observers successfully estimate the latent motion variable in each task. Further, the response statistics of the learned filters to naturalistic videos dictate that the optimal model for 3D speed estimation can be well-approximated by an extension of the widely studied energy model [1, 21]. Similar results are obtained for 3D direction estimation. Finally, for both 3D speed and direction estimation, we compare the performance of the two Bayesian observer models against previously reported human psychophysical data [4, 13, 23], and show that the models may provide a normative explanation of various aspects of human psychophysical performance.

## 2 Methods

### 2.1 Task design and dataset generation

A dataset of binocular video clips was generated for each 3D-motion task. In both tasks, a planar surface moved in 3D space, starting from 1 m straight ahead of the observer (Figure 1A,B). In the 3D speed estimation task, target surfaces moved towards or away from the observer with 3D speeds ranging from -2.5 m/s to 2.5 m/s in 0.125 m/s increments (Figure 1A). In the 3D direction task, the targets had fixed 3D speeds of 0.125 m/s, and moved along all directions in the XZ plane in 7.5° increments (Figure 1B). To generate naturalistic videos, the target surfaces were image patches sampled from a natural scene dataset [10]. In each binocular video, we applied a specific 3D motion to a sampled natural image patch. The 3D motion causes 2D motion on the retina of each eye, which we compute via projective geometry (Figure 1C). Left- and right-eye movies are generated by translating the retinal images with the appropriate 2D speed for each eye. The resulting stereo videos did not include monocular motion cues like looming, a limitation that will be addressed in future work.

Binocular videos had a duration of 250 ms (60 Hz) and subtended $1°$ of visual angle in each eye (30 pix/deg). This field of view, which was enforced by a raised cosine spatio-temporal window, approximates the information received by a receptive field in early visual cortex. The model also incorporated the optics of the human eye and basic retinal physiology (e.g. temporal response function of human cones; see [8]). A sample of white noise $\gamma$ was added to each video stimulus $s$. Each frame of the video was then vertically averaged to produce a 1D binocular video. Vertical averaging does not change the information available to vertically oriented receptive fields, because such receptive fields themselves vertically average the stimulus (see [8]). Finally, consistent with standard models of neural response [17, 18, 5, 2], the stimulus was contrast-normalized:

$$c = \frac{s + \gamma}{\|s + \gamma\|} \tag{1}$$

A given stereo movie obtained with this process is represented (for training or testing) as a vector with 900 elements (2 eyes x 30 pixels x 15 frames). For each 3D speed or 3D direction dataset, 500 naturalistic image movies were generated (Figure 1D).

## 2.2 Model structure and training

To obtain 3D motion estimates from the retinal videos, we applied a small set of linear filters to the videos, and decoded 3D motion from the filter responses using a probabilistic non-linear decoder. For each task, we learned linear filters (receptive fields) that maximize the accuracy of the estimates, using Accuracy Maximization Analysis (AMA) [9, 20, 16]. AMA has been used to train similar models for related visual tasks. Below, we describe the details of the model and training procedure.

**Stimulus encoding.** Filter responses are generated by filtering the contrast-normalized retinal input $c$, and adding a sample of white noise to the filter response:

$$R_i = \boldsymbol{f_i}^T \boldsymbol{c} + \eta \tag{2}$$

where $\boldsymbol{f_i}$ is a filter, $\eta \sim \mathcal{N}\left(0, \sigma_0^2\right)$ is a sample of filter response noise, and $R_i$ is the noisy response of filter $\boldsymbol{f_i}$ to the natural contrast-normalized stimulus $\boldsymbol{c}$. Denoting the filter set $\boldsymbol{f} = [\boldsymbol{f_1}, ..., \boldsymbol{f_n}]$ and population filter response $\boldsymbol{R} = [R_1, ..., R_n]$, the noisy filter responses to a particular stimulus $P(R_i|\boldsymbol{c}) \sim \mathcal{N}\left(\boldsymbol{f_i}^T \boldsymbol{c}, \sigma_0^2\right)$ are normally distributed.

**Stimulus decoding.** Each stimulus $\boldsymbol{s}$ is associated with a certain value $X_j$ (speed or direction) of the 3D motion latent variable $X$. The task involves estimating from a particular vector of filter responses $\boldsymbol{R}$ generated by stimulus $\boldsymbol{s}$, the value of the latent variable $X$ (speed or direction) associated with the stimulus. To obtain an optimal estimate (for a given cost function), the posterior probability $P(X = X_j|\boldsymbol{R})$ must be obtained for each value $X_j$, which can be computed as the product of the likelihood $L(X_j; \boldsymbol{R}) = P(\boldsymbol{R}|X_j)$ and the prior $P(X_j)$. For learning the filters, we compute the likelihood functions by approximating each conditional response distribution as a normal distribution, $P(\boldsymbol{R}|X_j) \sim \mathcal{N}(\boldsymbol{\mu_j}, \boldsymbol{\Sigma_j})$, where the conditional mean $\boldsymbol{\mu_j}$ and covariance $\boldsymbol{\Sigma_j}$ are estimated from the responses to the training set. Previous work on related tasks (binocular disparity estimation and 2D motion estimation) show that the filters learned with the assumption that the conditional response distributions are normal are almost identical to those learned without the assumption [20]. From the posterior probabilities, an estimate of the latent variable, and an associated cost (see below) can be obtained. We use the maximum a posteriori (MAP) estimator in the main text, although performance patterns are similar for other estimators (e.g. the minimum mean squared error estimator).

**Filter learning.** For learning the filters, we minimize cross-entropy loss, which is equivalent to minimizing the negative log-likelihood of the true value of the latent variable. Because we assume the conditional filter responses $P(\boldsymbol{R}|X)$ to be Gaussian–and because the empirically determined conditional filter responses are approximately Gaussian–the negative log-likelihood of the true value of the variable $X_k$ is given by a quadratic combination of the filter responses:

$$-LL(X_k; \boldsymbol{R}) = \frac{1}{2}(\boldsymbol{R} - \boldsymbol{\mu_k})^T \boldsymbol{\Sigma_k}^{-1} (\boldsymbol{R} - \boldsymbol{\mu_k})^T + B \tag{3}$$

where $B$ is a constant. Thus, we train the model by minimizing the expected negative log-likelihood across the dataset:

$$\underset{\boldsymbol{f}}{\arg\min} \, C = \frac{1}{N} \sum_{k,l} C_{k,l} \tag{4}$$

where the expected cost of an individual stimulus $l$ belonging to latent class $X_k$ is:

$$C_{k,l} = E_{\boldsymbol{R}_{k,l}} \left[ -ln P(X_k | \boldsymbol{R}_{k,l}) \right] \tag{5}$$

We use numerical methods to find the filters that minimize the loss function, under the constraint that each filter has unit magnitude. We trained eight filters for each model.

## 3 Results

### 3.1 3D speed estimation

The filters that optimize 3D speed estimation with naturalistic stimuli, the filter responses, and the resulting ideal observer estimates of 3D speed are shown in Figure 2A-D. The filters are Gabor-like and localized in spatio-temporal frequency. The filter responses cluster as a function of 3D speed. And the 3D speed estimates, which are decoded by the ideal observer from the filter responses, are accurate and precise.

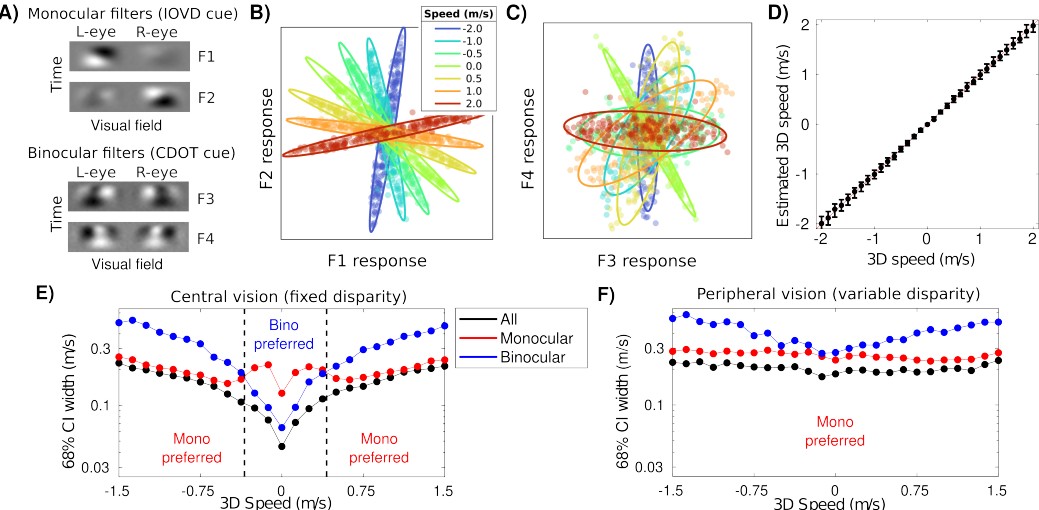

Figure 2: **3D speed estimation. A)** Four (of eight) learned filters for 3D speed estimation. Top: monocular filters (related to IOVD cue). Bottom: binocular filters (related to CDOT cue). **B)** Conditional filter responses for the pair of monocular filters in A, color coded by 3D speed (seven of 41 speeds are shown). Each dot is the response to an individual stimulus. Ellipses show the best-fitting Gaussians. **C)** Same as B but for the binocular filters in A. **D)** Median estimated 3D speeds as a function of true 3D speed. Error bars show 68% confidence intervals. **E)** Confidence intervals on estimates derived from monocular filters alone (red), binocular filters alone (blue), or all filters together (black). **F)** Same as E, but with variable initial fixation disparities to simulate increased disparity variability in the periphery.

The filters that optimize 3D speed estimation performance have some interesting properties. First, two distinct types of filters were spontaneously learned for the task. Half of the eight filters assign similar weights to each of the two eyes (binocular filters; Figure 2B). The other half assign strong weights to one eye and weights near zero to other eye (monocular filters; Figure 2B). The emergence of monocular filters is not trivial (PCA does not return monocular filters). The fact that monocular filters are learned may be surprising given that all filters received stereo videos as inputs. To understand why two filter types are learned, we examined how they support 3D speed estimation.

At slow 3D speeds, estimates derived exclusively from the binocular filters have smaller confidence intervals (i.e. are more precise) than the estimates obtained from monocular filters alone. At fast speeds, the opposite is true (Figure 2E). With all filters, the confidence intervals are more similar to those of the monocular filters at fast speeds, and more similar to those of the binocular filters at slow speeds. Thus, the two filter types are engaged in a division of labor. Each filter type has a domain of specialization. The filters extract complementary task-relevant information from the stimuli.

Interestingly, the two distinct filter types appear to be related to two different 3D motion cues used by humans and non-human primates in 3D motion estimation: the interocular velocity difference (IOVD) cue and the changing disparity over time (CDOT) cue. The IOVD cue first involves computing the velocity of each retinal image, and then computing the velocity differences between the eyes. The monocular filters are well-suited to this computation. The CDOT cue first involves a binocular comparison of the two retinal images at each time point to extract instantaneous disparity, and then computing how disparity changes in time. The binocular filters are well-suited to this computation.

Ideal observer performance is supported by the distinct filter types in a manner that dovetails with human psychophysical results in 3D speed discrimination tasks. Humans weight CDOT cues more heavily at low speeds and in the fovea, and IOVD cues more heavily at high speeds and in the peripheral visual field [13]. These similarities between how ideal and human observers estimate 3D speed suggest that the adaptive manner in which humans use these motion cues across conditions may reflect near-optimal processing of the available visual information.

Next, we examined whether the current framework could be used to account for differences between human foveal and peripheral 3D motion processing. One factor that might contribute to the dominance

of IOVD in human peripheral vision is that binocular disparities are more variable in the retinal periphery than in the fovea, This increased variability arises from binocular fixation patterns and from the depth structure of natural scenes [24, 19]. To approximate the stimulation received by the peripheral retinas, we generated a new dataset with variability in the initial disparity of the binocular videos; the original dataset had no initial disparity variability. Figure 2F shows that with increased disparity variability, the binocular filters (the CDOT cues) are less useful at all speeds than the monocular filters (the IOVD cues), including those where binocular filters originally dominated. (The decoder was matched to the filter response statistics with this new dataset.) This fact, along with the fact that the periphery tends to be stimulated by faster speeds, helps account for why the periphery relies near-exclusively on IOVD cues. We predict that filters learned from datasets having variable initial disparities will contain no binocular filters. If this prediction is confirmed, it will provide a normative justification for why human peripheral vision relies near-exclusively on IOVD cues.

There are additional points to make about how the filters respond to naturalistic stereo movies. First, the information about 3D speed is carried near-exclusively by the covariance of the filter responses. Second, the filter responses, conditioned on each 3D speed, are tightly approximated by Gaussian distributions (Figure 2B,C). These results is are important on methodological grounds because they justify the use of AMA-Gauss, which approximates each conditional response distribution $P(\boldsymbol{R}|X_j)$ as Gaussian during the filter learning process [20]. More fundamentally, these results indicate that quadratic combination of the filter responses (see Equation 3) is required for optimal inference of local 3D speed. These optimal computations are closely related to those posited by the energy model, a popular descriptive model of complex cells. Energy-model-like computations have previously been used to model neural selectivity for retinal speed [1, 8, 20, 11], disparity [22, 7, 14] and texture segmentation [3], and have been described as canonical neural computations [21]. The current results therefore provide a normative explanation, grounded in natural scene statistics, for the success of a common descriptive model of neural response.

## 3.2    3D direction estimation

The results for the 3D direction estimation task are similar in many ways to the 3D speed estimation results. The filters that optimize 3D direction estimation, their responses to naturalistic stimuli, and the ideal observer estimates of 3D direction are shown in Figure 3A-D. The filters are Gabor-like, the filter responses cluster as a function of 3D direction, and the estimates are largely accurate.

Two distinct types of filters are learned for the 3D direction estimation task, like with 3D speed estimation. One filter type (6 of 8) encodes the frontoparallel motion component but is blind to towards-away motion (frontoparallel filters; Figure 3A). The other (2 of 8) encodes the towards-away motion component, and is insensitive to frontoparallel motion (towards-away filters; Figure 3A).

Frontoparallel filters distinguish 3D motion directions by their frontoparallel motion component (Figure 3B). Towards-away filters primarily distinguish motion directions by the sign of the towards-away motion component (Figure 3C). Estimates derived exclusively from the 6 frontoparallel filters confuse the sign of towards-away motion approximately 50% of the time (Figure 3E, red). Estimates derived from the towards-away filters confuse the sign a small percentage of the time (Figure 3E, blue). The opposite pattern is exhibited when these estimates are evaluated with respect to the accuracy of the decoded frontoparallel motion component. Performance is good with frontoparallel filters and poor with towards-away filters (Figure 3F). The two types of filters have clear functional specializations. Again, like with 3D speed estimation, a large proportion of the task-relevant information is encoded in the covariance of the filter responses, and that the response distributions are reasonably approximated as Gaussian (Figure 3B,C). However, some filter responses are multi-modal (Figure 3B inset). This finding suggests that although energy-model-like computations extract most of the task-relevant information for local 3D direction estimation, more sophisticated computations are required for optimal decoding in natural images. The performance differences between quadratic decoding and optimal decoding remain to be determined.

Finally, ideal observer performance is similar to key aspects of human psychophysical performance, also like for 3D speed. For a non-negligible proportion of the stimuli, the ideal observer confuses towards and away motion directions (Figure 3G), generating a characteristic X-shaped pattern of responses. This surprising pattern of responses is characteristic of human performance in laboratory-based 3D motion tasks [4, 23] (Figure 3H), indicating that this type of error may be a consequence of optimal decoding of 3D direction.

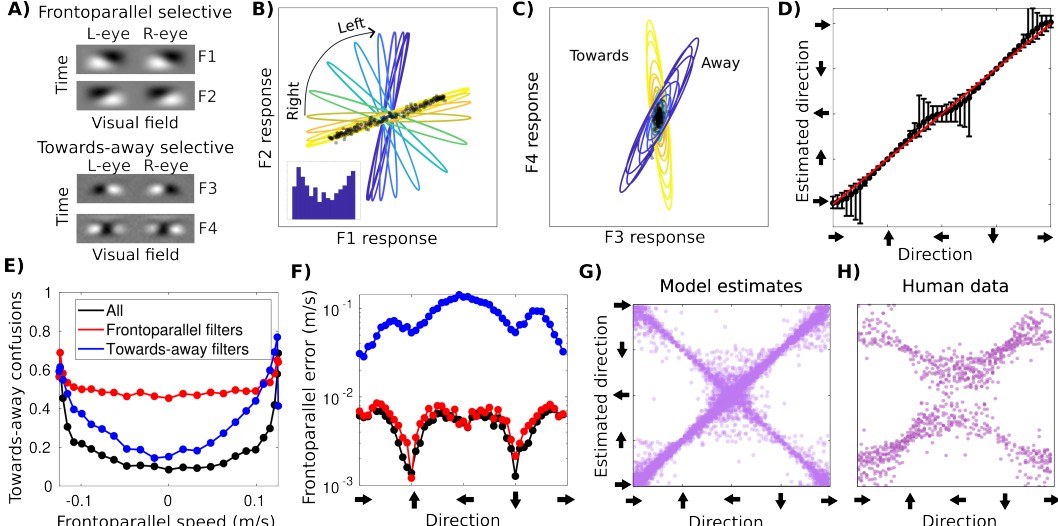

Figure 3: **3D direction estimation A)** Four (of eight) learned filters for 3D direction estimation. Top: Frontoparallel selective. Bottom: Towards-away selective. **B)** Conditional response distributions for the pair of left-right selective filters. Ellipses show Gaussians fitted to the filter responses conditional on each 3D direction, color coded by the value of the frontoparallel motion component. 3D directions with identical frontoparallel components but opposite towards-away components produce perfectly overlapping response distributions. Note that not all responses are perfectly Gaussian distributed (black dots inset; see main text). **C)** Same as B, but for towards-away filters, color coded by towards-away motion component. **D)** Estimated 3D directions. Error bars are 68% confidence intervals. **E)** Proportion of towards-away sign confusions obtained from the different types of filters. Confusion rates of over 0.5 are possible for some motions because the sign could be towards, away, or none (e.g. frontoparallel motion). **F)** Mean absolute error of the frontoparallel component of estimated motion as a function of true 3D motion direction. **G)** Model estimates across the dataset. Each point shows the estimate from a unique stimulus. **H)** Same as G for human data (adapted from [4]).

## 4  Conclusions

We developed ideal observers from naturalistic stereo video clips for the tasks of 3D speed and 3D direction estimation. The current findings suggest a normative explanation for various counter-intuitive properties of human 3D motion processing and estimation performance, and deepen our understanding of why the energy model has proven a useful model of neural response. In the 3D speed estimation task, the ideal observer learns both monocular and binocular filters, and makes adaptive use of these two distinct filter types in a manner similar to how humans use IOVD and CDOT 3D motion cues [13]. In the 3D direction estimation task, two distinct filter types are also learned and the patterns of ideal observer performance are again quite similar to the patterns produced by humans (e.g. towards-away confusions) in laboratory settings [4, 23]. Although the filter types, which are specialized for discriminating frontoparallel and towards-away motion components, have no reported analogs in the scientific literature to our knowledge, their emergence from the current analyses should motivate future neurophysiological and psychophysical studies to test for their existence and effects. Another clear direction for future work is the development of a single ideal observer model for the joint task of simultaneously estimating both 3D speed and 3D direction. Although there may be surprises, we anticipate that many of the findings will generalize. Regardless, the present results suggest that multiple aspects of human 3D motion processing may result from optimal computations in the early visual system.

**Acknowledgements**

This work was supported by the National Eye Institute and the Office of Behavioral and Social Sciences Research, National Institutes of Health Grant R01-EY028571 to J.B.

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
