# OpenReview forum: "Image-computable Bayesian model for 3D motion estimation with natural stimuli explains human biases"
_NeurIPS.cc/2022/Workshop/SVRHM — SVRHM Poster_

### Official Review · Reviewer_fZYf · 2022-10-13
**Image-computable Bayesian model for 3D motion estimation with natural stimuli explains human biases**

**Rating:** 6
**Confidence:** 2

**Review:**


This work presents a Bayesian approach to model 3D motion estimation. More specifically the paper proposes two tasks to estimate motion having speed and direction as latent variables for each task respectively. The authors train these models by learning filters to reproduce human response taken from a reference.


Pros

The problem to imitate human response to 3D motion through ML models seems novel. The work shows interesting conclusions about how the human estimation reflect an optimized computation for 3D motion estimation, what type of Bayesian model and assumed response distribution fits better which task, and that an energy-model-like performs better in the 3D speed estimation task but not in 3D direction one.


Cons

It is not clear how the generated binocular movies dataset is similar to the one used to report human psychophysical data in the specified references, and neither the exact input from which filters extract stimulus features. Besides, it is not clear if the monocular filters are learned separated from the binocular filters or all in one model. Is the input a pair of images for binocular and one image for monocular, are they sequences of frames to simulate the motion/video?

At the beginning of section Results, it is mentioned that Figure 2A shows 4 out 8 binocular filters, however the caption of the image mentions they are monocular and binocular. Some more explanation about monocular and binocular experiments, as well as stimulus is highly recommended.

The manuscript does not present enough review of the state of the art, nor explicitly mention how recent is this idea and how few works has approached to it. Are there other Bayesian Models perhaps design for monocular/binocular data but other types of movies? Do other works usually model motion through these two tasks?

The manuscript does not explain or hypothesize why do energy-model-like computations might not be optimal for the 3D direction estimation task.

The authors do not support or explain how the selected response model is tightly linked to the properties of the primate visual system.

Minor:
It is suggested that every section has a brief introduction of how subsections interact
Correct small typos in writing, i.e. is the optimal computation for 3D speed estimation, but not 3D direction estimation (but not for?).

---

### Official Review · Reviewer_qXGU · 2022-10-14
**More technical details and comparisons are required**

**Rating:** 6
**Confidence:** 4

**Review:**

In this paper, the authors propose a Bayesian observer model for motion speed and direction estimation from images. They use an algorithm called Accuracy Maximization Analysis (AMA) for estimating a set of 2d filters that are read out with a nonlinear function to estimate motion speed and direction. It is interesting to see that the trained model was able to reproduce some aspects of psychophysics from humans. However, the paper requires more additional details about the methodology used in the training process.

Major comments:

1- Based on the explanations in section 2.2, it seems that the loss function is the log-likelihood of the training data conditioned on the filters responses. Is that correct? Though, I can't understand where the nonlinearity entered into the optimization process. It was mentioned a few times throughout the paper that a quadratic nonlinearity was used for reading out from the filters, but its exact form wasn't shown in the equations.

2- Why did the authors choose the AMA approach for optimization? How would the results be different if, instead, another optimization algorithm was used, such as gradient descent? In the absence of comparisons with other methods, it is impossible to judge whether any of the observations (e.g. figures 3 and 4) are specific to the proposed model, or other optimization approaches would lead to similar results. Also, the paper could benefit from more details about the AMA method.

Minor comment:

1- The references in the main text should be modified to include publication year.
2- How does violating the assumption of equal mean and variance of filter outputs influence the results? The filter responses were assumed to be "Poisson-like" (equal mean and variance) but they were actually modeled with a Gaussian distribution. I assume the AMA method could not handle filter outputs with Poisson distribution?

---

### Official Review · Reviewer_UjM9 · 2022-10-16
**Great work with interesting findings, but has few overblown claims**

**Rating:** 8
**Confidence:** 2

**Review:**

The paper presents an ideal observer model for estimating motion speed and direction from a binocular view of a moving surface. Model is trained finds an optimal linear encoder by maximizing accuracy on motion estimation tasks of a Bayesian decoder operating on the encoded representations. The recovered features reflect those related to changing disparity and interocular velocity difference cues, and the behavior of the ideal observer model is qualitatively similar to those of humans.

One concern with regards to the paper is that some terms such as "naturalistic" and "optimal" are often used liberally, without the proper qualifications.

- The model presented in the paper is not necessarily an "optimal visual information processing model", but an ideal observer model that learns the optimal linear encoders whose features are decoded optimally for the specific tasks. For instance, there may be other, complex nonlinear encoder-decoders that performs better on the task. This also does not also prove the optimality or non-optimality of quadratic computations (sections 3.1, 3.2). The constraints and assumptions are more informative about an information processing system and the representations it carries than the claim that it is optimal.
-  What makes the set of stimuli used in the study "naturalistic"? How were the natural image textures generated? Is the use of natural image textures the most important in a motion velocity estimation? (e.g. We may use shape information, information about occlusion scenes and other visual cues in naturalistic environments). What does this stimuli achieve that artificial random dot patterns cannot?

Other comments:
- Section 1 paragraph 3: why does non-Gaussian responses imply non-optimality?
- Fig 1C: wrong figure on the bottom? The top row and the bottom row show the same features. Which direction on the y-axis does time flow?
- Fig 2a: axes labels for the filters? (x and time?)
- Fig 2,3: a scalar term "speed" seem more accurate than the vector term "velocity"
- Section 3.1 paragraph 1 - typo on the posterior formula